# Femicide in Mexico: Statistical evidence of an increasing trend

**Eva Selene Hernández Gress** [1], **Martin Flegl** [2]*, **Aleksandra Krstikj** [3], **Christina Boyes** [4]

**1** School of Engineering and Sciences, Tecnologico de Monterrey, Pachuca, Hidalgo, Mexico, **2** School of Engineering and Sciences, Tecnologico de Monterrey, Mexico City, Mexico, **3** School of Architecture, Art and Design, Tecnologico de Monterrey, Campus Estado de México, Cd López Mateos, Mexico, **4** Centro de Investigación y Docencia Económicas, A. C., División de Estudios Internacionales (International Studies Division), Ciudad de México, México

* martin.flegl@tec.mx

**Data Availability Statement:** All relevant data are located at https://doi.org/10.6084/m9.figshare.21224915.v1.

## Abstract

This study analyzes whether femicide in Mexico has increased more severely than other life and bodily integrity crimes (e.g., homicide, culpable homicide, injuries, malicious injuries, abortion, and other crimes that threaten life). To achieve this, the Executive Secretariat of the National Public Security System database was cleaned and the number of femicides per 100,000 inhabitants was calculated, for the period from January 2016 to March 2022 in all states of Mexico. Through descriptive statistics, non-parametric analysis of means, and hypothesis tests, we demonstrate that the states with the highest number of femicides are the Estado de Mexico (State of Mexico), Ciudad de Mexico (Mexico City), and Veracruz; moreover, the number of femicides exhibits a growing trend while the total number of life and bodily integrity crimes does not. Finally, we forecast the number of femicides for the next five months. To our knowledge, there is no other article that analyzes the growth trend of femicide compared to other crimes. Visualizing and understanding that femicide is on the rise compared with other types of crimes can help the government and legislators generate policies that are consistent with the magnitude of the problem.

## Introduction

In 1979, the UN Convention on the Elimination of All Forms of Discrimination Against Women (CEDAW) was created. As of 2001, all countries in Latin America had signed and ratified the convention [1]. CEDAW spawned a series of domestic reforms within Latin America to address femicide as a criminal act distinguishable from other forms of homicide. Additionally, the Belem do Para Convention, adopted in 1994, is an exclusively Latin American human rights instrument designed to draw attention to violence against women in the region and cultivate strategies to protect women and defend their rights to live free of violence. The convention has been ratified by all but 5 states in Latin America and the Caribbean, and the five that have not yet ratified have acceded to the convention [2].

Despite international and domestic laws enshrining women's right to a life free from violence and numerous campaigns to define and decrease violence against women, levels of violence against women in Latin America are the highest in the world [3]. The Mexican case is of

**Funding:** The authors received no specific funding for this work.

**Competing interests:** The authors have declared that no competing interests exist.

particular interest due to the culture of "machismo" which aggravates violence against women and femicide by tacitly supporting male dominance in the home and the use of violence against partners and spouses [4–6].

In September 2020, Mexican feminists drew international attention to the problem of gender-based violence and femicides (the terms femicide and feminicide are used interchangeably throughout the text) in Mexico when they took over a federal human rights office [7]. Mexico is recognized as having one of the highest rates of femicides internationally, a trend many suggest is increasing. Ten women per day die because of femicide in this Latin American country [8]. Within academia, gender-based violence in Mexico has drawn interdisciplinary attention, attracting researchers from across the social sciences as well as urban planning engineers, public health experts, and policymakers. The presence and severity of the increase in femicides in Mexico remains a question of debate for politicians, policymakers, and the society. There is a debate in Mexico involving the current president Andres Manuel Lopez Obrador who has argued that both femicide and homicide stem from the neoliberal period, during which family disintegration and the loss of values occurred, leading to an increase in overall violence [9]. Due to the importance of the problem and the controversy surrounding it, scholars are also interested in determining if an increase in Mexican femicides exists.

There are several reasons to suspect an increase in femicides in Mexico. Worldwide, femicide is increasing, a problem that was aggravated by the coronavirus pandemic. Latin America has shown particularly sharp increases in femicide. In 2020, Mexico's rate of femicide did not increase according to data from the Gender Equality Observatory, yet a shockingly high number of women were victims of the crime, with 948 out of every 100,000 Mexican women dying of femicide. Of countries in Latin America, only Brazil exhibited a higher rate of femicide (Fig 1). Furthermore, the National System of Statistical and Geographical Information (INEGI) of Mexico recently released a report demonstrating an increase in violence against women more broadly in Mexico [10]. The women's movement in Mexico has also been very vocal regarding what they perceive to be an increasing trend in femicides, and the international community has voiced concern regarding the level of femicide in Mexico [11]. What remains to be seen is if the data agrees with public and international impressions and overall trends in data regarding violence against women in the country, i.e., the purpose of this paper.

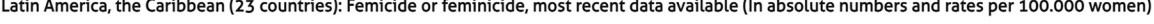

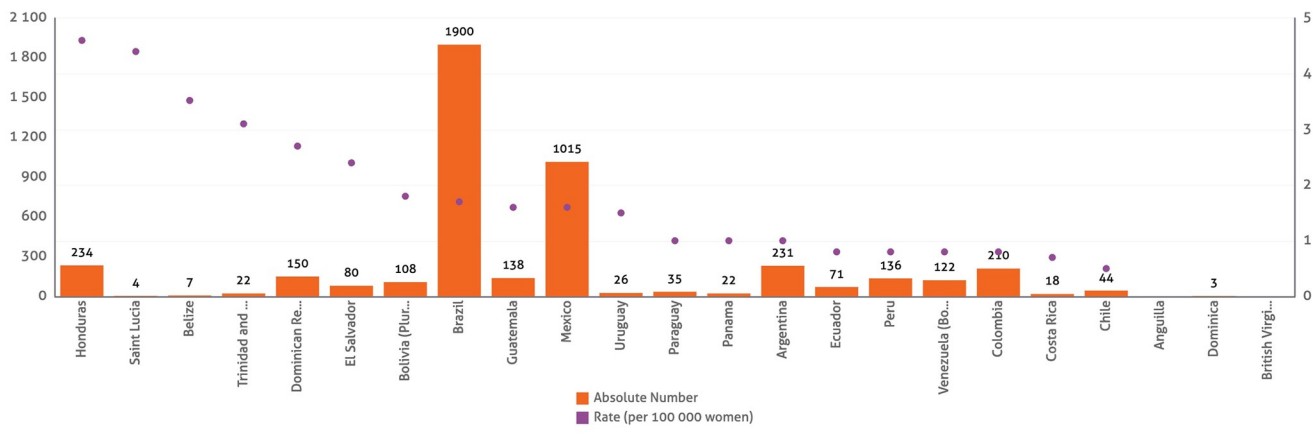

Latin America, the Caribbean (23 countries): Femicide or feminicide, most recent data available (In absolute numbers and rates per 100.000 women)

Source: ECLAC, Gender Equality Observatory for Latin America and the Caribbean

**Fig 1. Femicide in 21 Latin America and Caribbean states.** Source: ECLAC [12].

As of 2015, 93% of defendants in domestic violence cases in Mexico were male and the majority of victims were female [13]. Additionally, the percentage of women murdered as a proportion of total people murdered has been growing for more than a decade. According to Charles-Leija et al. [14], drawing on data from the Mexican government, the proportion of women murdered increased from 1.4 to 4.6 of all homicides between 2007 and 2012. This figure represents a staggering growth rate of more than 200 percent.

## Literature review

Beginning in 2015, several studies discuss trends in femicides in different countries. Some, such as Frias [15], only use descriptive statistics. Karakasi et al. [16] analyze the trends in femicide in Greece for the period 2010–2021 finding that in 2021, while homicides decreased to 89 incidents per year, domestic homicides skyrocketed to 34 cases per year, reaching the highest annual number ever recorded nationally. Frias [15] argues that femicides in Mexico are increasing, with indigenous groups showing the greatest growth. Molinatti and Acosta [17] analyzed the growth trend of femicides in women who were abused in several Latin American countries, finding that in Mexico the number of femicides among women aged 15 to 59 increased between 2001 and 2011.

In more quantitatively rigorous studies, Ramos de Souza et al. [6] and Cardoso Meira et al. [18] analyze the homicide mortality of Brazilian women using negative binomial regression, the former in the last 35 years, and the latter only in the states of the Northwest region between 1980 and 2014. Torrecilla et al. [19] proposed a statistical methodology to identify temporal interferences in the count of time series and test their methodology with a study of femicides in Spain from 2007 to 2017. The authors find that femicides decreased during the period studied. Valdez Rodríguez et al. [20] who use system dynamics to simulate the behavior of femicide in Mexico conclude that it will continue to grow exponentially.

Curious as to the actual number of femicides and trends in the data on violence against Mexico's women, we set out to answer two key questions. (RQ1) Have the number of femicides in Mexico increased since 2015? (RQ2) Can we attribute the increase to an overall increase in violent crimes or is the trend in femicides independent?

The rest of the article is organized as follows. In materials and methods: the data, method, and fundamental concepts are presented. In the analysis, the results were explained considering descriptive statistics and non-parametric means comparison; also, a hypothesis proportion test is performed between the years and a forecast to describe the trend; at the end of the section the discussion is displayed. Finally, the conclusion and further research is described.

## Materials and methods

### Data

This analysis uses data related to the criminal incidents in Mexico from the Executive Secretariat of the National Public Security System (SESNSP) [21]. The criminal incidence refers to the presumed occurrence of crimes recorded in preliminary investigations initiated or investigation files, reported by the Attorney General's Office and the Attorney General's Office of the states.

The dataset covers a period from January 2016 until March 2022 (87 months) across all 32 Mexican states. The registered crimes are divided into seven juridical types: *Property crimes*; *Family crimes*; *Sexual freedom and safety crimes*; *Society crimes*; *Life and bodily integrity crimes*; *Personal freedom crimes*; and *Other affected legal assets (of common jurisdiction)*.

## Method

To answer the research questions, the analysis was carried out in four different stages:

1. *Descriptive statistics*—Analysis of the number of femicides in the states, where the mean and standard deviation were obtained, analyzing the change by state in the years 2015 to 2021.

2. *Comparison* of whether the average femicides and the *Life and bodily integrity crimes* are the same during the 2015–2021 period.

3. *Hypothesis test* to confirm if the proportion of femicides with respect to the total of crimes is increasing.

4. *Forecasting methods* to shed light on the future trend of femicides in Mexico.

## Descriptive statistics

Graphs are useful for understanding data trends, however more formal analysis requires the calculation and interpretation of the numerical measures' summary, which serve to characterize the dataset and communicate the salient features. Measures of concentration include the mean and the median.

The mean for a data set $x_1, x_2, \ldots x_n$, which is the most widely used measure of concentration is simply the average of the data [22] and is given by Eq 1.

$$\bar{x} = \frac{\sum_{i=1}^{n} x_i}{n} \tag{1}$$

The median, on the other hand, is the mean value once the observations are ordered from smallest to largest.

$$\tilde{x} = \left\{ n \text{ is odd} = \left(\frac{n+1}{2}\right)^{th}, n \text{ is even} = \text{average of } \left(\frac{n}{2}\right)^{th} \text{and} \left(\frac{n}{2}+1\right)^{th} \right\} \tag{2}$$

Variability means show how far the data is from its mean; the variance measures show how much average deviation there is from the data with respect to the mean [22], as can be seen in the following equation.

$$s^2 = \frac{\sum_{i=1}^{I} (x_i - \bar{x})^2}{n-1} \tag{3}$$

The standard deviation $s$ is the root of the variance $s^2$ and it has the same units of the data.

## Means comparison

The simplest analysis of variance (ANOVA) is known as one-way or one-way ranking and involves analysis of data sampled from more than two numerical populations (distributions) or of data from experiments in which more than two treatments were used. The characteristic that differentiates the treatments of the populations is called the factor under study and the different treatments or populations are known as factor levels.

One-factor ANOVA focuses on the comparison of more than two populations or treatment means [23]. Let $I$ be the number of populations or treatments being compared and $\mu_i$ the population mean $i$ or the true mean response when treatment $i$ is applied ($i = 1, 2, \ldots, I$).

The relevant hypotheses are $H_o$: $\mu_1 = \mu_2 = \cdots = \mu_I$ against $H_a$: at least two of the $\mu_i$ are different. If $H_o$ is true if all $\mu_i$ are identical, whereas $H_o$ is rejected when at least one pair is different.

If the sample sizes are assumed to be equal, let $J$ be the number of observations in each sample, the data set is $IJ$. The means of each sample will be denoted by $\bar{X}_1$, $\bar{X}_2$, ... $\bar{X}_I$, hat is,

$$\bar{X}_{i.} = \frac{\sum_{j=1}^{J} X_{ij}}{J}, i = 1, 2 \ldots, I \tag{4}$$

The dot in the second subscript means that all the values of that subscript were added together while holding the value of the other subscript fixed. Likewise, the average of all observations is called the large mean.

$$\bar{X}.. = \frac{\sum_{i=1}^{I} \sum_{j=1}^{J} X_{ij}}{IJ} \tag{5}$$

Further, let $S_1^2, S_2^2, \ldots, S_I^2$ be the sample variances, where

$$S_i^2 = \frac{\sum_{j=1}^{J} \left( X_{ij} - \bar{X}_{i.} \right)^2}{J - 1}, i = 1, 2, \ldots, I \tag{6}$$

Each observation $X_{ij}$ within any sample is assumed to be independent of one another. The distributions of population $I$ are considered normal with the same variance. That is, each $X_{ij}$ is normally distributed with

$$E\left( X_{ij} \right) = \mu_i \text{ and } V\left( X_{ij} \right) = \sigma^2 \tag{7}$$

If the assumption of normality and equal variances between the samples are not met, other non-parametric tests such as the Kruskal-Wallis test or the Mood test can be applied [22]. In the case of the Kruskal-Wallis test, let $N = \sum_{i=1}^{I} J_i$ be the total number of observations in the data set, and suppose that all $N$ observations are ordered from 1 (the smallest $X_{ij}$) to N (the largest $X_{ij}$). When $H_o$: $\mu_1 = \mu_2 = \cdots = \mu_I$ is true, all observations come from the same distribution, in which case all possible assignments of the ranks 1,2,...$N$ to the $I$ samples are equally likely, and the ranks are expected to be mixed in these samples. However, if $H_o$ is false, some samples will be made up of observations that have small ranges in the pooled sample, while others will be made up of observations with large ranges.

Formally if $R_{ij}$ denotes the range of the $X_{ij}$ between the $N$ observations and $R_{i.}$ and. $\bar{R}_{i.}$ denote, respectively, the total and the average of the ranges of the $i$–$th$ sample, so when $H_o$ is true:

$$E\left( R_{ij} \right) = \frac{N + 1}{2}, \text{ and } E(\bar{R}_{i.}) = \frac{1}{J_i} \sum_{i=1}^{I} E\left( R_{ij} \right) = \frac{N + 1}{2} \tag{8}$$

The Kruskal-Wallis test is a measure of the magnitude to which the $\bar{R}_{i.}$ deviate from their common expected value $\frac{N+1}{2}$, and is given by Eq 9

$$K = \frac{12}{N(N + 1)} \sum_{i=1}^{I} J_i \left( \bar{R}_{i.} - \frac{N + 1}{2} \right)^2 \tag{9}$$

The values of $K$ at least as contradictory to $H_o$ are the $k$ values that are equal to or exceed $K$. This is an upper tail test, that is, the value of $P = P_0(K \geq k)$. Under $H_o$, each possible assignment of the ranks of the $I$ samples are equally likely. So, in theory, all assignments can be

enumerated, the value of $K$ for each determined, and the null distribution obtained by counting the number of times that each of the $K$ values are present. When $H_o$ is true and $I = 3$, $J_i \leq$ 6, $i = 1,2,3$ or $I > 3$, $J_i \geq 5$, $i = 1,2,\ldots,I$, then $K$ has a chi-square distribution with $I − 1$ degrees of freedom. This implies that a $P$ value is approximately equal to the area under the curve $\chi^2_{I-1}$ to the right of $k$. This implies that if this value of $k$ is less than the level of significance $\alpha$, $H_o$ will be rejected and at least one pair of means will be different, where *reliability* $= 1 − \alpha$.

## Two proportions hypothesis test

An individual is considered successful $S$ in a population if it possesses some characteristic of interest. Then the proportions in two different samples can be defined as:

- $p_1$ = proportion of successes $S$ in a population 1.

- $p_2$ = proportion of successes $S$ in a population 2.

That is, $p_1$ is the probability that an individual is successful in the population. Suppose that a sample size $n_1$ is selected in the first population and $n_2$ in the second. Let $X$ be the number of successes $S$ in the first sample and $Y$ be the number of successes in the second. The independence of $X$ and $Y$ is assumed, whenever the two sample sizes are much smaller than the corresponding population, the distributions of $X$ and $Y$ are binomial. The natural estimator of the difference of the proportions $p_1 − p_2$ is the corresponding difference between the sample proportions $\frac{X}{n_1} − \frac{Y}{n_2}$.

Devore [23] shows that if $\hat{p}_1 = X/n_1$ and $\hat{p}_2 = Y/n_2$ where $X \sim Bin(n_1, p_1)$ and $Y \sim Bin(n_2, p_2)$ with $X$ and $Y$ as independent variables, then the expected value is

$$E(\hat{p}_1 − \hat{p}_2) = p_1 − p_2 \tag{10}$$

so that $\hat{p}_1 − \hat{p}_2$ is an unbiased estimator of $p_1 − p_2$ and the variance is

$$V(\hat{p}_1 − \hat{p}_2) = \frac{p_1 q_1}{n_1} + \frac{p_2 q_2}{n_2}, \ where \ q_i = 1 − p_i \tag{11}$$

So, if the distributions of $\hat{p}_1, \hat{p}_2$ are approximately normal, the distribution of the estimator $\hat{p}_1 − \hat{p}_2$ can be assumed to be approximately normal. By standardizing $\hat{p}_1 − \hat{p}_2$ a variable $Z$, whose distribution is approximately standard normal, is obtained.

$$Z = \frac{\hat{p}_1 − \hat{p}_2 − (p_1 − p_2)}{\sqrt{\frac{p_1 q_1}{n_1} + \frac{p_2 q_2}{n_2}}} \tag{12}$$

The most general null hypothesis would be the form $H_o: p_1 − p_2 = \Delta_0$, $\Delta_0 = 0$ and $\Delta_0 \neq 0$ must be considered separately. The real problems are mostly of the case $\Delta_0 = 0$, that is $H_o: p_1 = p_2$. When $H_o: p_1 = p_2$ is true, let $p$ be the combined ratio of $p_1$ and $p_2$ and $= 1 − p$. So, the standardized variable is:

$$Z = \frac{\hat{p}_1 − \hat{p}_2 − (0)}{\sqrt{pq\left(\frac{1}{n_1} + \frac{1}{n_2}\right)}} \tag{13}$$

This variable has approximately a normal distribution when $H_o$ is true. By replacing $p$ and $q$ with appropriate estimators a test statistic is obtained. If $p = p_1 = p_2$ instead of separate samples of size $n_1$ and $n_2$ from two different population with binomial distributions, we have a single sample of size $n_1 + n_2$ from a population with proportion $p$. The total number of individuals in

this pooled sample that have the characteristic of interest $X + Y$. The natural estimator of $p$ is therefore:

$$\hat{p} = \frac{X + Y}{n_1 + n_2}, \ \hat{p}_1 = \frac{X}{n_1}, \ \hat{p}_2 = \frac{Y}{n_2} \tag{14}$$

Then $\hat{p}$ is a weighted average of the estimators $p_1$ and $p_2$ with the following formulation:

Null hypothesis: $H_o: p_1 - p_2 = 0$

Statistical value of test $\frac{Z = \hat{p}_1 - \hat{p}_2}{\sqrt{pq\left(\frac{1}{n_1} + \frac{1}{n_2}\right)}}$

| Alternative hypotheses | Determination of $p - value$ |
|---|---|
| a) $H_i: p_1 - p_2 > 0$ | Area under the standard normal curve to the right of the $z$ |
| b) $H_i: p_1 - p_2 < 0$ | Area under the standard normal curve to the right of the $z$ |
| c) $H_i: p_1 - p_2 \neq 0$ | $2*$(area under the standard normal curve to the right of $z$) |

The security test is that $\hat{p}_1 n_1, \hat{q}_1 n_1, \ \hat{p}_2 n_2, \hat{q}_2 n_2$ is at least 10.

## Forecasting methods

There are different prediction methods; those that are used in a time series from historical values are known as extrapolation methods. In a forecasting method by extrapolation, it is assumed that past patterns and trends will continue into future months, without considering what caused the past data, they simply assume the trends and patterns will continue.

Although there are some characteristics such as trend or seasonality that help to prefer one forecast method over another, the fact is that it is accuracy measures such as MAD (Mean Absolute Deviation), MAPE and MSD that help to choose one method over another.

MAD is the measurement of the error size in units, the formula is:

$$MAD = \frac{\sum |Y_i - \hat{Y}_i|}{n} \tag{15}$$

$Y_i$ is the real value of the series at the period $i$

$\hat{Y}_i$ is the forecasting in the period $i$

$n$ is the number of periods

MSE maximizes the error by raising the square of the differences, punishing those whose difference was higher compared to others, being suitable for periods with small deviations.

$$MSE = \frac{\left(Y_i - \hat{Y}_i\right)^2}{n} \tag{16}$$

MAPE measures the deviation in percentage terms and not in units like the previous measures. It is the average of the absolute error or difference between the actual demand and the forecast, expressed as a percentage of the actual value.

$$MAPE = \frac{\sum \frac{|Y_i - \hat{Y}_i|}{Y_i}}{n} \tag{17}$$

Single exponential smoothing is one of the most widely used methods of forecasting because it requires little computation. This method is recommended when the data has no trend, that is, there is no cyclical variation or pronounced trend. In contrast, double exponential smoothing is a method used when the data has a trend [24]. In this case two parameters are required: level and trend. The level serves to smooth the estimation of the data and the trend calculates an average growth at the end of the period [25]. The formula is:

$$\hat{Y}_{t+m} = \alpha Y_t + (1 - \alpha)\hat{Y}_t \tag{18}$$

$$a_t = 2S_t - S_t'$$

$$b_t = \frac{\alpha}{1 - \alpha}(S_t - S_t')$$

$$S_t = \alpha Y_t + (1 - \alpha)S_{t-1}$$

$$S_t' = \alpha S_t + (1 - \alpha)S_{t-1}'$$

$S_t$ Is the smoothed value $Y_t$ in the time $t$

$S_t'$ Is the double smoothed value $Y_t$ in the time $t$

$a_t$ Calculates the difference between exponentially smoothed values

$b_t$ Is the adjustment factor

$\hat{Y}_{t+m}$ Is the forecast $m$ periods ahead.

## Analysis and discussion

In this section, we present the data analysis and discuss the results.

### Descriptive statistics

As mentioned previously, SESNSP [21] dataset covers a period from January 2016 until March 2022 (87 months) across 32 Mexican states. The registered crimes are divided into seven juridical types: Property crimes; Family crimes; Sexual freedom and safety crimes; Society crimes; Life and bodily integrity crimes; Personal freedom crimes; and Other affected legal assets (of common jurisdiction).

During the analyzed period, 13,814,735 crimes were registered; out of these 1,809,735 cases (13.10%) belong to the Life and bodily integrity crimes (homicide, culpable homicide, injuries, malicious injuries, abortion, and other crimes that threaten life). In this juridical type of crime, 5,759 femicide cases were registered (with standard deviation 2.66), which represents 66.20 registered cases per month (per 100,000 habitants) in the whole of Mexico. The data is in the repository 10.6084/m9.figshare.22111211. Table 1 summarizes the descriptive statistics of the registered crimes for all 32 Mexican states.

Código Penal Federal [26] article 325 defines femicide as a crime when someone deprives a woman of her life for reasons of gender. Gender reasons are considered to exist when any of the following circumstances occur I. The victim shows signs of sexual violence of any kind; II. Infamous or degrading injuries or mutilations have been inflicted on the victim, before or after the deprivation of life or acts of necrophilia; III. There is a history or data of any type of violence in the family, work, or school environment, of the active subject against the victim; IV. There has been a sentimental, affective or trust relationship between the perpetrator and the victim; V. There is evidence that establishes that there were threats related to the criminal

**Table 1. Descriptive statistics of monthly registered Life and bodily integrity crimes in Mexico per state and year.**

| Year | | Total | Life and bodily integrity crimes | Homicide | Injuries | Femicide | Abortion | Others |
|------|------|-------|----------------------------------|----------|----------|----------|----------|--------|
| 2015[a] | Max | 28,637.00 | 5,954.00 | 356.00 | 5,648.00 | 10.00 | 17.00 | 84.00 |
| | Min | 0.00 | 17.00 | 2.00 | 2.00 | 0.00 | 0.00 | 0.00 |
| | Mean | 4,317.20 | 645.02 | 83.33 | 550.84 | 0.91 | 1.45 | 8.30 |
| | StDev | 5,048.70 | 962.30 | 65.91 | 916.59 | 1.82 | 2.57 | 15.04 |
| 2016[b] | Max | 31,514.00 | 6,426.00 | 286.00 | 6,097.00 | 11.00 | 17.00 | 89.00 |
| | Min | 0.00 | 12.00 | 3.00 | 2.00 | 0.00 | 0.00 | 0.00 |
| | Mean | 4,588.10 | 620.89 | 92.46 | 515.16 | 1.58 | 1.47 | 10.22 |
| | StDev | 5,222.06 | 850.02 | 69.61 | 804.54 | 2.14 | 2.28 | 16.75 |
| 2017 | Max | 32,498.00 | 5,959.00 | 326.00 | 5,594.00 | 13.00 | 11.00 | 85.00 |
| | Min | 157.00 | 18.00 | 3.00 | 7.00 | 0.00 | 0.00 | 0.00 |
| | Mean | 5,050.77 | 657.91 | 107.62 | 533.55 | 1.93 | 1.42 | 13.39 |
| | StDev | 5,647.84 | 881.79 | 78.91 | 828.07 | 2.50 | 2.12 | 19.55 |
| 2018 | Max | 30,474.00 | 5,421.00 | 414.00 | 5,058.00 | 16.00 | 15.00 | 122.00 |
| | Min | 164.00 | 24.00 | 3.00 | 5.00 | 0.00 | 0.00 | 0.00 |
| | Mean | 5,182.11 | 655.61 | 115.85 | 517.02 | 2.34 | 1.57 | 18.83 |
| | StDev | 5,811.87 | 824.24 | 89.43 | 757.96 | 2.80 | 2.57 | 25.18 |
| 2019 | Max | 32,089.00 | 6,045.00 | 416.00 | 5,650.00 | 15.00 | 18.00 | 185.00 |
| | Min | 186.00 | 19.00 | 7.00 | 7.00 | 0.00 | 0.00 | 0.00 |
| | Mean | 5,393.66 | 691.55 | 116.82 | 548.30 | 2.47 | 1.87 | 22.10 |
| | StDev | 5,955.60 | 920.23 | 90.81 | 851.10 | 2.85 | 3.25 | 27.65 |
| 2020 | Max | 31,768.00 | 5,951.00 | 444.00 | 5,503.00 | 19.00 | 19.00 | 216.00 |
| | Min | 128.00 | 12.00 | 4.00 | 6.00 | 0.00 | 0.00 | 0.00 |
| | Mean | 4,794.76 | 617.97 | 112.67 | 473.96 | 2.47 | 1.65 | 27.22 |
| | StDev | 5,488.17 | 823.08 | 95.02 | 748.05 | 2.92 | 2.93 | 34.82 |
| 2021 | Max | 35,218.00 | 6,153.00 | 356.00 | 5,689.00 | 19.00 | 22.00 | 238.00 |
| | Min | 147.00 | 25.00 | 5.00 | 6.00 | 0.00 | 0.00 | 0.00 |
| | Mean | 5,323.26 | 675.25 | 114.49 | 523.31 | 2.54 | 1.83 | 33.07 |
| | StDev | 6,159.25 | 890.26 | 87.91 | 815.98 | 2.95 | 3.23 | 42.75 |
| 2022[c] | Max | 36,313.00 | 5,966.00 | 354.00 | 5,473.00 | 17.00 | 23.00 | 194.00 |
| | Min | 316.00 | 26.00 | 5.00 | 10.00 | 0.00 | 0.00 | 0.00 |
| | Mean | 5,300.23 | 654.82 | 105.79 | 514.40 | 2.39 | 2.16 | 30.09 |
| | StDev | 6,193.37 | 870.82 | 83.93 | 795.81 | 2.86 | 4.02 | 38.91 |
| | TOTAL | 13,814,368 | 1,809,735 | 111,670 | 1,450,720 | 5,759 | 4,518 | 53,938 |

[a] Incomplete data for 8 months in Oaxaca

[b] Incomplete data for1 month in Oaxaca

[c] only 3 months are included for all states

act, harassment, or injuries from the subject against the victim; VI. The victim has been held incommunicado, regardless of the time prior to the deprivation of life; VII. The body of the victim is exposed or exhibited in a public place. In the case that femicide is not proven, rules of homicide are applied.

## Means comparison

We analyzed whether there was a significant difference between femicides per state during the period of study; thus, 32 samples were considered. This analysis was not possible with ANOVA because the samples did not meet the normality assumptions and the variances were

not the same. The Kolmogorov-Smirnov test was performed to test normality, and we obtained a $p < 0.001$ for every sample. Thus, we concluded that the samples do not meet the assumption of normality as S1 Fig indicates. As a result, the test of variances was not performed.

Two non-parametric tests were performed: the Kruskall Wallis and Mood Tests, which are shown in Tables 2 and 3, respectively- During the evaluated period, an overall median 1.0 case of femicides per 100,000 habitants was reported monthly. The results indicate statistically significant differences between the state medians ($p < 0.000$). The highest number of femicides is reported in Estado de México with a median of 7.5 registered cases, followed by Veracruz (6),

**Table 2. Kruskal-Wallis test: Femicide versus state.**

| States | N[a] | Median | Mean Rank | Z-Value |
|---|---|---|---|---|
| Aguascalientes | 84 | 0.0 | 649.1 | -8.32 |
| Baja California | 84 | 1.0 | 1317.6 | -0.27 |
| Baja California Sur | 84 | 0.0 | 575.0 | -9.21 |
| Campeche | 84 | 0.0 | 708.4 | -7.60 |
| Chiapas | 84 | 2.0 | 1674.8 | 4.03 |
| Chihuahua | 84 | 1.0 | 1282.9 | -0.69 |
| Ciudad de México | 84 | 4.0 | 2159.6 | 9.87 |
| Coahuila de Zaragoza | 84 | 1.0 | 1299.7 | -0.48 |
| Colima | 84 | 0.0 | 945.9 | -4.74 |
| Durango | 84 | 0.0 | 871.4 | -5.64 |
| Guanajuato | 84 | 1.0 | 1350.1 | 0.12 |
| Guerrero | 84 | 1.0 | 1309.9 | -0.36 |
| Hidalgo | 84 | 1.0 | 1309.8 | -0.36 |
| Jalisco | 84 | 4.0 | 2098.2 | 9.13 |
| México | 84 | 7.5 | 2452.5 | 13.39 |
| Michoacán de Ocampo | 84 | 1.5 | 1423.3 | 1.00 |
| Morelos | 84 | 2.0 | 1617.9 | 3.35 |
| Nayarit | 84 | 0.0 | 752.1 | -7.08 |
| Nuevo León | 84 | 4.0 | 1748.9 | 4.92 |
| Oaxaca | 75 | 3.0 | 1956.9 | 7.01 |
| Puebla | 84 | 2.0 | 1620.4 | 3.38 |
| Querétaro | 84 | 0.0 | 835.8 | -6.07 |
| Quintana Roo | 84 | 0.0 | 948.3 | -4.72 |
| San Luis Potosí | 84 | 2.0 | 1386.5 | 0.56 |
| Sinaloa | 84 | 3.0 | 1788.8 | 5.40 |
| Sonora | 84 | 2.0 | 1751.3 | 4.95 |
| Tabasco | 84 | 2.0 | 1471.1 | 1.58 |
| Tamaulipas | 84 | 0.0 | 832.3 | -6.11 |
| Tlaxcala | 84 | 0.0 | 707.6 | -7.61 |
| Veracruz de Ignacio de la Llave | 84 | 6.0 | 2366.4 | 12.36 |
| Yucatán | 84 | 0.0 | 734.9 | -7.28 |
| Zacatecas | 84 | 1.0 | 998.9 | -4.11 |
| Overall | 2679 | | 1340.0 | |
| H = 1185.89 | DF = 31 | P < 0.000 | | |
| H = 1256.22 | DF = 32 | P < 0.000 | (Adjusted for ties) | |

[a] represents one month or period

**Table 3. Mood median test: Femicide versus state.**

| States | Median | N < = Overall Median | N > Overall Median | Q3 –Q1 | 95% Median CI |
|---|---|---|---|---|---|
| Aguascalientes | 0.0 | 82 | 2 | 0.00 | (0, 0) |
| Baja California | 1.0 | 51 | 33 | 2.00 | (1, 1.42679) |
| Baja California Sur | 0.0 | 81 | 3 | 0.00 | (0, 0) |
| Campeche | 0.0 | 80 | 4 | 1.00 | (0, 0) |
| Chiapas | 2.0 | 24 | 60 | 2.75 | (2, 3) |
| Chihuahua | 1.0 | 45 | 39 | 3.00 | (0, 2) |
| Ciudad de México | 4.0 | 7 | 77 | 3.75 | (4, 5) |
| Coahuila de Zaragoza | 1.0 | 50 | 34 | 1.75 | (1, 2) |
| Colima | 0.0 | 68 | 16 | 1.00 | (0, 1) |
| Durango | 0.0 | 74 | 10 | 1.00 | (0, 1) |
| Guanajuato | 1.0 | 48 | 36 | 1.00 | (1, 2) |
| Guerrero | 1.0 | 45 | 39 | 2.00 | (1, 2) |
| Hidalgo | 1.0 | 47 | 37 | 2.00 | (1, 2) |
| Jalisco | 4.0 | 8 | 76 | 3.75 | (4, 5) |
| México | 7.5 | 0 | 84 | 7.75 | (6, 10) |
| Michoacán de Ocampo | 1.5 | 42 | 42 | 2.00 | (1, 2) |
| Morelos | 2.0 | 31 | 53 | 2.00 | (2, 3) |
| Nayarit | 0.0 | 81 | 3 | 1.00 | (0, 0) |
| Nuevo León | 4.0 | 29 | 55 | 6.00 | (2, 5) |
| Oaxaca | 3.0 | 14 | 61 | 3.00 | (3, 4) |
| Puebla | 2.0 | 31 | 53 | 3.00 | (2, 3) |
| Querétaro | 0.0 | 72 | 12 | 1.00 | (0, 0.426788) |
| Quintana Roo | 0.0 | 63 | 21 | 1.75 | (0, 1) |
| San Luis Potosí | 2.0 | 39 | 45 | 2.75 | (1, 2) |
| Sinaloa | 3.0 | 22 | 62 | 4.00 | (2, 3.42679) |
| Sonora | 2.0 | 20 | 64 | 2.00 | (2, 3) |
| Tabasco | 2.0 | 38 | 46 | 2.00 | (1, 2) |
| Tamaulipas | 0.0 | 71 | 13 | 1.00 | (0, 0) |
| Tlaxcala | 0.0 | 77 | 7 | 1.00 | |
| Veracruz de Ignacio de la Llave | 6.0 | 2 | 82 | 4.00 | |
| Yucatán | 0.0 | 80 | 4 | 1.00 | |
| Zacatecas | 1.0 | 66 | 18 | 1.00 | |
| Overall | 1.0 | | | | |
| Null hypothesis | H₀: The population medians are all equal | | | | |
| Alternative hypothesis | H₁: The population medians are not all equal | | | | |
| DF | Chi-Square | | | P-Value | |
| 31 | 1000.54 | | | 0.000 | |

Ciudad de México (4), Jalisco (4), and Nuevo León (4). Even though, Ciudad de México, Jalisco, and Nuevo León have a median of 4 femicides, Ciudad de Mexico has 77 periods above the median, Nuevo León 55, and Jalisco 37 (Table 2). The lowest femicide rates are observed in Baja California Sur (0, Aguascalientes (0), Campeche (0), Nayarit (0), Tamaulipas (0), Tlaxcala (0)), and Yucatán (0). Fig 2 presents the level of femicide across the country.

In the second part of the analysis, the means of *femicides* from January 2015 to December 2021 were compared to investigate the tendency of the reported cases, the year 2022 was excluded from the analysis as the data includes only 3 months of the year. To better understand trends in the baseline levels of violence, we also examined the tendency regarding the number

FEMICIDE RATE IN MEXICAN STATES 2015-2022

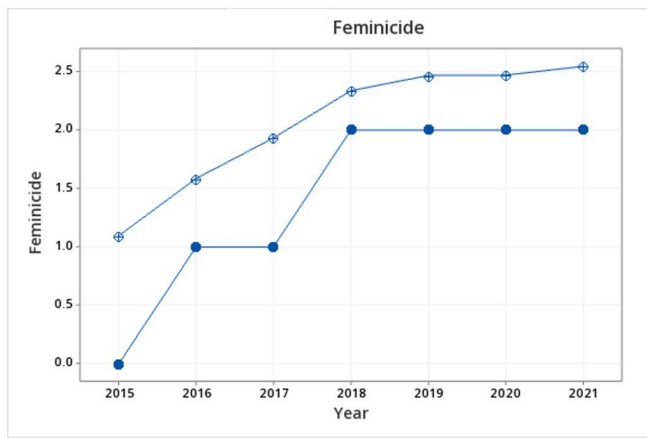

**Fig 2. Femicides per 100,000 habitants, means between 2015–2022.** Authors' elaboration.

of *life and bodily integrity crimes*. Fig 3 shows the comparison of the mean and median of both variables. The results reveal that the mean number of registered *femicides* has an increasing tendency, while the total crimes against life remain constant from 2015 to 2021. *Femicide* increased from an average of 83.53 per 100,000 inhabitants in 2015 to 181.03 cases in 2021, whereas 645 cases of *life and bodily integrity crimes* were reported in 2015 and 675.2 in 2021.

The normality for each year of femicides and *life and bodily integrity crimes* was analyzed. As S2 Fig indicates, normality was not fulfilled, and the assumption of equality of variances was not met. For this reason, non-parametric tests were used to formalize the statistical analysis.

Tables 4 and 5 present the results of the Kruskal-Wallis and Mood media tests for the reported cases of femicide with respect to each year. In both cases, $H_o$ is rejected ($p < 0.000$), i.e., there are differences between the years of the analysis. Both tests confirm the increasing tendency of femicide in Mexico, and the growth of the reported cases is statistically significant.

We analyzed whether a similar trend can be observed in the case of the reported numbers of *life and bodily integrity crimes*. In Tables 6 and 7, the same tests were applied. In this case, in

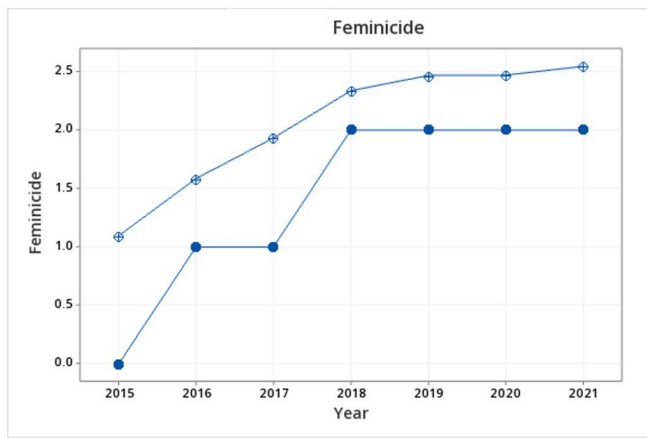
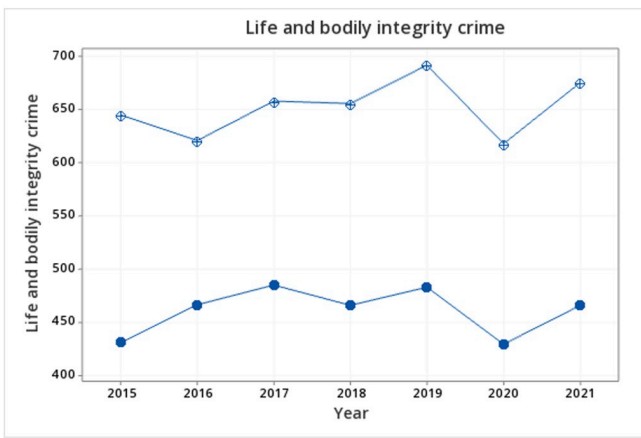

**Fig 3. Mean and median for femicide and life and bodily integrity crimes.** Authors' elaboration.

**Table 4. Kruskal-Wallis test: Femicide versus year.**

| Year | N | Median | Ave Rank | Z |
|---|---|---|---|---|
| 2015 | 376 | 0.0000 | 987.9 | -9.52 |
| 2016 | 383 | 1.0000 | 1,181.8 | -4.32 |
| 2017 | 384 | 1.0000 | 1,285.6 | -1.49 |
| 2018 | 384 | 2.0000 | 1,448.4 | 2.97 |
| 2019 | 384 | 2.0000 | 1,482.7 | 3.91 |
| 2020 | 384 | 2.0000 | 1,483.7 | 3.93 |
| 2021 | 384 | 2.0000 | 1,502.3 | 4.44 |
| Overall | 2,679 | - | 1,340.0 | - |
| H = 146.62 | DF = 6 | P < 0.000 | | |
| H = 155.31 | DF = 6 | P < 0.000 | (Adjusted for ties) | |

**Table 5. Mood median test: Femicide versus year.**

| Chi-Square = 87.77 | | DF = 6 | P < 0.000 | | |
|---|---|---|---|---|---|
| | | | | | Individual 95.0% CIs |
| Year | N ≤ | N ≥ | Median | Q3-Q1 | +---------+---------+---------+------ |
| 2015 | 277 | 99 | 0.00 | 2.00 | * |
| 2016 | 239 | 144 | 1.00 | 2.00 | (---------------* |
| 2017 | 224 | 160 | 1.00 | 3.00 | * |
| 2018 | 185 | 199 | 2.00 | 3.00 | (--------------* |
| 2019 | 187 | 197 | 2.00 | 2.00 | (--------------* |
| 2020 | 189 | 195 | 2.00 | 2.00 | (--------------* |
| 2021 | 187 | 197 | 2.00 | 2.75 | (--------------* |
| | | | | | +---------+---------+---------+------<br>0.00     0.60      1.20      1.80 |
| Overall median = 1.00 | | | | | |

the Kruskal-Wallis and Mood median tests, the *p*-value is greater than 0.05 (*p* = 0.097 and *p* = 0.070 respectively). Therefore, $H_o$ is rejected, that is, there are no statistically significant differences between the years in *life and bodily integrity crimes*. In other words, while in Tables 4 and 5 there are statistically significant changes in the number of *femicides*, in the total number of *life and bodily integrity crimes* no statistically significant trend is observed.

**Table 6. Kruskal-Wallis test: Life and bodily integrity crimes versus year.**

| Year | N | Median | Ave Rank | Z |
|---|---|---|---|---|
| 2015 | 376 | 431.5 | 1,276.1 | -1.73 |
| 2016 | 383 | 467.0 | 1,291.5 | -1.33 |
| 2017 | 384 | 485.5 | 1,351.8 | 0.32 |
| 2018 | 384 | 466.5 | 1,365.0 | 0.68 |
| 2019 | 384 | 483.5 | 1,401.9 | 1.69 |
| 2020 | 384 | 430.0 | 1,292.9 | -1.29 |
| 2021 | 384 | 466.5 | 1,399.4 | 1.63 |
| Overall | 2,679 | - | 1,340.0 | - |
| H = 10.72 | DF = 6 | P = 0.097 | | |
| H = 10.72 | DF = 6 | P = 0.097 | (Adjusted for ties) | |

**Table 7. Mood median test: Life and bodily integrity crimes versus year.**

| Chi-Square = 11.67 | | DF = 6 | P = 0.070 | | |
|---|---|---|---|---|---|
| | | | | | Individual 95.0% CIs |
| Year | N $\leq$ | N $\geq$ | Median | Q3-Q1 | ---+---------+---------+---------+--- |
| 2015 | 208 | 158 | 432 | 353 | (----------*------) |
| 2016 | 186 | 197 | 467 | 351 | (----------*------) |
| 2017 | 181 | 203 | 486 | 444 | (--------*-----) |
| 2018 | 188 | 196 | 467 | 442 | (--------*--------) |
| 2019 | 181 | 203 | 484 | 488 | (-------*-----------) |
| 2020 | 212 | 172 | 430 | 416 | (------*------) |
| 2021 | 187 | 197 | 467 | 429 | (-------*------) |
| | | | | | ---+---------+---------+---------+--- |
| | | | | | 400        440        480        520 |
| Overall median = 460 | | | | | |

## Hypothesis proportion test

The tests in the previous section indicated a significant difference in the reported *femicides* from 2015 to 2021, and no significant difference in the case of total number of *life and bodily integrity crimes* during the same period. However, we felt it was necessary to assess whether femicides were growing or decreasing related to *life and bodily integrity crimes* in the time horizon. For this, the following hypothesis was proposed using the test statistic of Eq 13, this equation is based on the assumption of normality in the Central Limit Theorem (CLT). According to the CLT, when *n* is large the probability $P(a < X < b)$ can be calculated assuming that the distribution is normal and standardize it.

$$H_0 : p_1 = p_2$$

$$H_1 : p_1 < p_2$$

$p_1$ = proportion of *femicides* related to *life and bodily integrity crimes* in a year.
$p_2$ = proportion of *femicides* related to *life and bodily integrity crimes* in the following year.

Table 8 summarizes the results of the analysis. The proportion of the reported cases of *femicides* with respect to the total number of *life and bodily integrity crimes* has generally increased over the evaluated period, although it remained constant in a few periods. The growth of the registered cases of *femicides* was statistically higher in 2016 compared to 2015 ($p < 0.000$), in 2017 compared to 2016 ($p = 0.005$), in 2018 compared to 2017 ($p < 0.000$) and in 2020 compared to 2019 ($p = 0.007$), whereas the period from 2018–2019 and from 2020–2021 do not demonstrate statistically significant increases ($p = 0.492$ and $p = 0.901$ respectively). The tendency is clearly visible in Fig 4.

The year 2020 in Fig 4 is interesting. There are studies that discuss the 2020 situation related to the pandemic in terms of people being less vulnerable to crimes because of reduced mobility in public spaces. However, violence against women worsened considerably in this period as many women spent prolonged confinement periods with their aggressors, not only in Mexico but in other Latin American countries as well [27].

## Forecasting

Finally, to make a forecast of the number of femicides that could be expected in Mexico in 2023, double exponential smoothing technique was used to analyze the time series. This

**Table 8. Test of proportions of the number of femicides related to life and bodily integrity crimes, comparing the time horizon.**

| $p_1(2015), p_2(2016)$ | | | | $p_1(2016), p_2(2017)$ | | | |
|---|---|---|---|---|---|---|---|
| Test and CI for Two Proportions | | | | Test and CI for Two Proportions | | | |
| Sample | X | N | Sample $p$ | Sample | X | N | Sample $p$ |
| 1 | 412 | 242,525 | 0.0017 | 1 | 607 | 237,800 | 0.0026 |
| 2 | 607 | 237,800 | 0.0026 | 2 | 742 | 252,638 | 0.0029 |
| Estimate for difference = -0.000853778 | | | | Estimate for difference = -0.000384444 | | | |
| 95% upper bound for difference: -0.000634949 | | | | 95% upper bound for difference: -0.000138826 | | | |
| Test for difference = 0 (vs < 0):<br>$Z = -6.43, p < 0.000$ | | | | Test for difference = 0 (vs < 0):<br>$Z = -2.57, p = 0.005$ | | | |
| $p_1(2017), p_2(2018)$ | | | | $p_1(2018), p_2(2019)$ | | | |
| Test and CI for Two Proportions | | | | Test and CI for Two Proportions | | | |
| Sample | X | N | Sample $p$ | Sample | X | N | Sample $p$ |
| 1 | 742 | 252,638 | 0.0029 | 1 | 897 | 251,756 | 0.0036 |
| 2 | 897 | 251,756 | 0.0036 | 2 | 947 | 265,555 | 0.0036 |
| Estimate for difference = -0.000625965 | | | | Estimate for difference = -0.00000314257 | | | |
| 95% upper bound for difference: -0.000634949 | | | | 95% upper bound for difference: 0.000269542 | | | |
| Test for difference = 0 (vs < 0):<br>$Z = -3.91, p < 0.000$ | | | | Test for difference = 0 (vs < 0):<br>$Z = -0.02, p = 0.492$ | | | |
| $p_1(2019), p_2(2020)$ | | | | $p_1(2020), p_2(2021)$ | | | |
| Test and CI for Two Proportions | | | | Test and CI for Two Proportions | | | |
| Sample | X | N | Sample $p$ | Sample | X | N | Sample $p$ |
| 1 | 947 | 265,555 | 0.0036 | 1 | 948 | 237,302 | 0.0040 |
| 2 | 948 | 237,302 | 0.0040 | 2 | 977 | 259,295 | 0.0038 |
| Estimate for difference = -0.000428793 | | | | Estimate for difference = 0.000227000 | | | |
| 95% upper bound for difference: -0.000143192 | | | | 95% upper bound for difference: 0.000517744 | | | |
| Test for difference = 0 (vs < 0):<br>$Z = -2.49, p = 0.007$ | | | | Test for difference = 0 (vs < 0):<br>$Z = 1.29, p = 0.901$ | | | |

approach is used when the series exhibits a trend, but not a seasonal pattern (as happens with femicides). Further, considering all the data but the decreasing weights in older observations, each month in each year is considered as a period of the time series (87 periods). Table 9 presents the forecasted expected femicide cases and the double exponential smoothing, and

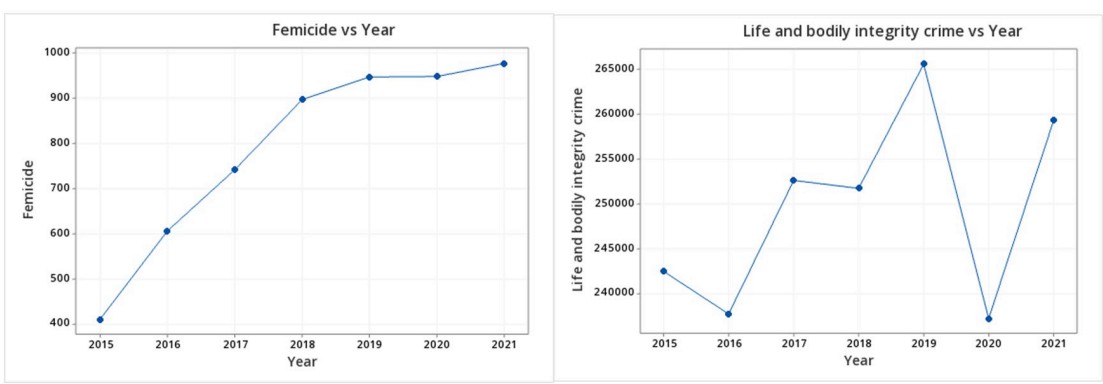

**Fig 4. Variation of femicides and life and bodily integrity crimes (per year).** Author's elaboration.

**Table 9. Forecast of expected femicide cases in Mexico for April-August 2023.**

| Period | Year | Forecast | Lower | Upper |
|--------|------|----------|-------|-------|
| 88 | April | 73.6522 | 52.1342 | 95.170 |
| 89 | May | 73.7804 | 47.3444 | 100.216 |
| 90 | June | 73.9086 | 42.1015 | 105.716 |
| 91 | July | 74.0369 | 36.6001 | 111.474 |
| 92 | August | 74.1651 | 30.9411 | 117.389 |

confidence intervals are shown in Fig 5. It can be observed that a growth of femicide cases is to be expected. Comparing the forecast against the actual data, the forecast is even lower, but the forecast is within the confidence interval [28].

## Discussion

Between 10 and 11 women are murdered every day in Mexico, while women, girls, and adolescents are kidnapped, and many are sexually violated throughout the country [29]. Within the framework of International Women's Day, the National Citizen Observatory of Femicide (OCNF), a national network made up of 43 organizations and located in 23 Mexican states, warned of its concern about the prevalence of femicide in Mexico. We asked if femicide is part of a general trend of increasing violence in the country or if femicide is increasing independent of other crimes against life. Although newspaper articles show that femicide is growing [30–32], to our knowledge, no post-2010 research article shows that femicides are growing faster than other non-typical crimes against life as such. Additionally, we were only able to identify one scientific article that used quantitative data and techniques to describe femicide in Mexico [20]. Thus, the objective of this article was to answer whether the number of femicides per 100,000 inhabitants is increasing compared to all crimes against life in Mexico.

Our results demonstrate that there is an increase in the number of femicides across most years of our analysis, except for 2018–2019 and 2020–2021. Furthermore, by employing non-

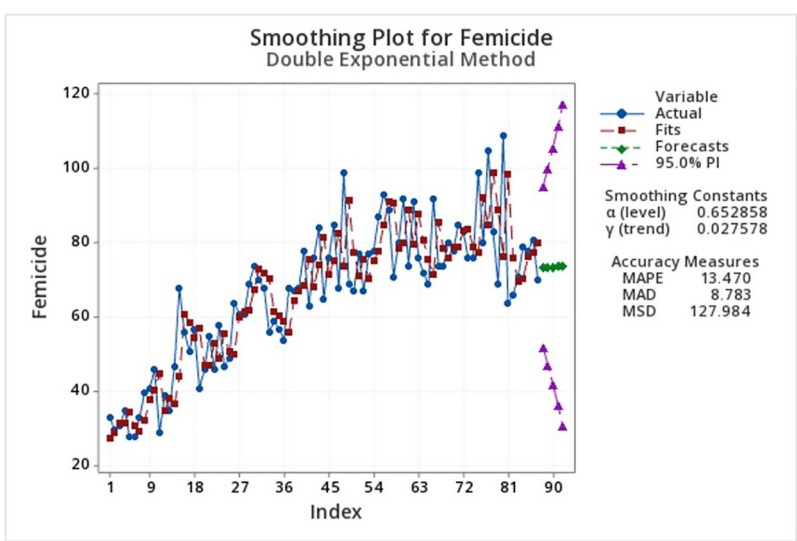

**Fig 5. Forecast for femicides in the next five months.** Authors' elaboration.

parametric means tests we found that there was a difference between the years for femicide (RQ1), but not for crimes against life (RQ2).

To assess whether femicides grew at a higher rate than the total number of crimes against life during our period of study, we carried out an analysis of proportions of the number of femicides with respect to the total number of crimes against life. We found that the proportion is decreasing in some periods and constant in others, which shows that femicides are growing at a different rate than the total number of crimes against life. In other words, the tendencies observed in femicide are categorically different than for other crimes against life and bodily harm.

The findings of our research set the foundation for researchers to examine what social, political, and economic factors are most relevant to femicide in Mexico, and to reevaluate public policies in Mexico regarding femicide.

## Conclusion

Since the 90's, laws were to protect women from violence and numerous campaigns were launched to promote respect for women. Nevertheless, femicide in Mexico has increased. Mexico has one of the highest femicide rates in Latin America and worldwide.

Mexican policymakers debate whether the trend of femicide is greater than other crimes against life if femicide warrants attention as a specific and separate area of concern that policymakers should seek to directly address. To shed light on this question, we addressed, two research questions and collected data from the Executive Secretariat of the National Public Security System. This database was cleaned to describe the number of femicides and other crimes against life per 100,000 inhabitants in all states of Mexico from January 2016 to March 2022.

Using non-parametric tests (Kruskal Wallis and Mood) in addition to hypothesis tests, we found that the Mexican states with the highest number of femicides are the Estado de México, Ciudad de México, and Veracruz. Additionally, we found that the average number of femicides has increased since 2015 (RQ1) while that of other crimes against life has not (RQ2).

In publishing this research, we hope to draw policymakers' attention to femicide as a priority and to raise social awareness of the gravity of the problem. Furthermore, we recognize that this problem is multifactorial and plan future studies addressing the factors which impact the growth of femicides in Mexico from a multidisciplinary perspective.

## Supporting information

**S1 Fig. Test of normality of the annual data of registered femicide per state, in all cases such as $p \leq 0.01$, $H_0$ is rejected, that is, the data is not normally distributed Baja California, Guanajuato, México and Nuevo León are presented as examples.** Authors' elaboration. (TIF)

**S2 Fig. Test of normality of the annual data of registered femicide and total crimes against life and bodily integrity, in all cases such as $p \leq 0.01$, $H_0$ is rejected, that is, the data is not normally distributed.** Authors' elaboration. (TIF)

## Author Contributions

**Conceptualization:** Eva Selene Hernández Gress, Martin Flegl, Aleksandra Krstikj, Christina Boyes.

**Data curation:** Martin Flegl.

**Formal analysis:** Eva Selene Hernández Gress, Christina Boyes.

**Investigation:** Eva Selene Hernández Gress, Aleksandra Krstikj, Christina Boyes.

**Methodology:** Eva Selene Hernández Gress, Martin Flegl, Christina Boyes.

**Software:** Eva Selene Hernández Gress.

**Visualization:** Aleksandra Krstikj.

**Writing – original draft:** Eva Selene Hernández Gress, Martin Flegl, Aleksandra Krstikj, Christina Boyes.

**Writing – review & editing:** Christina Boyes.

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
