## [Decision Letter · Decision Letter 0]

9 Feb 2023

PONE-D-22-26945Femicide in Mexico: Evidence of an increasing trendPLOS ONE

Dear Dr. Flegl,

Thank you for submitting your manuscript to PLOS ONE. After careful consideration, we feel that it has merit but does not fully meet PLOS ONE’s publication criteria as it currently stands. Therefore, we invite you to submit a revised version of the manuscript that addresses the points raised during the review process.

We look forward to receiving your revised manuscript.

Kind regards,

Jesús Espinal-Enríquez

Academic Editor

PLOS ONE

Journal Requirements

This research did not receive any specific grant from funding agencies in the public, commercial, or not-for-profit sectors. 

Additional Editor Comments:

Dear Dr. Flegl,

First of all, I want to sent my apologies for the delay in your manuscript assessment. It has been very difficult to find two reviewers with a proper expertise to revise your manuscript.

Regarding your submission PONE-D-22-2694, after the evaluation of the Reviewers' comments and also my own revision, I consider that the manuscript cannot be accepted in its current version, and it is necessary a major correction of it. Despite that one of the reviewers have considered to reject this submission, I guess that the concerns raised by the Reviewer can be addressed in a second version of the manuscript. The Reviewer's comments are mainly focused on the statistical meaning of your results and the assumptions made of your data. These concerns must be clearly addressed with a proper statistical testing.

The comments of Reviewer 2 concerning the "paradox of information" also should be broadly commented in your manuscript and as a response to the Reviewer's evaluation.

Reviewers' comments:

Reviewer's Responses to Questions

**Comments to the Author**

1. Is the manuscript technically sound, and do the data support the conclusions?

Reviewer #1: No

Reviewer #2: Yes

2. Has the statistical analysis been performed appropriately and rigorously? 

Reviewer #1: No

Reviewer #2: Yes

3. Have the authors made all data underlying the findings in their manuscript fully available?

Reviewer #1: No

Reviewer #2: Yes

4. Is the manuscript presented in an intelligible fashion and written in standard English?

Reviewer #1: Yes

Reviewer #2: Yes

5. Review Comments to the Author

Reviewer #1: COMMENTS

Femicide in Mexico: Evidence of an increasing trend

General remarks

1. The title of the article does not explain the topic correctly, if a statistical analysis was carried out then it should be indicated explicitly. What “evidence” of an increasing trend has been found?

2. The definition of crimes against life should be pointed out explicitly in the abstract. Against which other crimes are femicides compared to in the article?

3. The phrase “We understand that all femicides are not reported or that there are cases that are not adequately classified” is important, since definitely this is a problem. Why point out that femicides are not adequately reported or classified? Better definitions or doing research on the topic is definitely needed at this point.

4. In general, it is interesting to compare data of different periods of time. This should be treated carefully though, since temporal data of femicides in Mexico is very scattered across time; because of this fact we suggest revising the whole aim of the paper once again.

Mistakes in methodology, writing and numbering mistakes

1.Throughout the paper, assumptions of normality and independence are made in the data, but this is not well supported throughout the text. It is necessary to establish clearly why it can be thought that the phenomenon complies with normality and independence.

2. Page 5 the authors specify the numbering of the sections.

The rest of the article is organized as follows. In section 2, materials and methods were presented, including descriptive statistics of the data and the method used for the analysis. In section 3, the results were explained considering parametric and non parametric means comparison, also a hypothesis proportion test was performed between the years. Section 4 presents the results and in section 5 we discussed them. Finally, the conclusion and further research is described in section 6.

However, this numbering is not present in the titles, it is necessary to put the numbering of the sections and subsections if they exist.

3. On page 5:

Table 1 summarizes the descriptive statistics of the registered crimes for all 32 Mexican states.

This corresponds to a result. The descriptive analysis is presented in the Method section in point 1. This table should appear in the results and not in the Data section.

4. The results section begins with the ANOVA results, this corresponds to point 2 (page 6) of the three stages of the analysis. This section should begin with point 1 of the analysis stages (Descriptive statistics).

5. The graphs should be improved, for example on page 16 in one of the graphs the years appear at the bottom and in the other at the top. It must standardize the presentation of the graphs and improve the quality.

6. The forecast part does not appear in the analysis stages nor in the methods section, it should be presented clearly in the methods, what double exponential smoothing consists of as they did with the other methods.

7. On page 13, the tables to which it refers must be 4 and 5, since they are the ones that describe the statistical significance of the number of femicides.

“In other words, while in Tables 4 and 5 there are statistically significant changes in the number of femicides, in the total number of life and bodily integrity crimes no statistically significant trend is observed”. In the case of all crimes against life and bodily integrity, if it corresponds to tables 6 and 7.

1. In the ANOVA analysis, the variability of the behavior of femicides in the period of time is very large, if we take the average 2.0692 cases of femicides per 100,000 inhabitants and the standard deviation 1.9265, we can verify the great variation in the data by calculating the coefficient of variation of 93%.

If we analyze the columns of the mean and the standard deviation of Table 2 (page 10), we observe some states with not very high coefficients of variation, which were listed with a high number of femicides, except for the state of Morelos:

State Mean StDev Cv

Ciudad de México 4.786 2.742 57.3

Jalisco 4.464 2.595 58.13

Estado de México 8.548 4.433 51.8

Morelos 1.1711 2.345 49.94

Veracruz 6.524 3.198 49.02

The variability of states is not so high with respect to the others, so it will be necessary to evaluate whether these states meet the normal conditions. Since if fulfilled, the femicides in these states will be intrinsic natural variabilities, that is, characteristics of the states. And in the other states the dynamics will be non-linear to the presence of femicides.

General evaluation

Despite the fact that the topic is very important, the paper is in many ways incomplete and very poorly written.

Reviewer #2: The manuscript analyzes an issue of special interest in Mexico: whether the practice of femicide has increased icompared to other equivalent crimes. The study is based on official information from the Executive Secretariat of the National Public Security System from which the evidence presented is drawn. Although the evidence supports the hypothesis that femicides have increased in the country, it is considered relevant for the co-authors to review their analytical framework, specifically to reflect on the paradox of information in human rights (Sikkink), a situation that occurs when a practice is named and begins to be counted. While the evidence presented is compelling, it is important to develop reasons why their analysis would go some way to controlling the information paradox. On the other hand, given that it does not make the central argument of the paper, it is suggested to avoid the reference to the relationship between the approval of legislation and the increase in crime as well as to the macho culture.

6. PLOS authors have the option to publish the peer review history of their article (what does this mean?). If published, this will include your full peer review and any attached files.

Reviewer #1: No

Reviewer #2: No

---

## [Author Response · Author response to Decision Letter 0]

21 Mar 2023

To whom may it concerns:

We thank the reviewers for their time and dedication in the reviewing process. In response to the suggestions and comments provided, we revised the manuscript. The reviewers’ comments are included below, as well as the steps we took to address each suggestion. 

Reviewers' comments:

Reviewer's Responses to Questions

Comments to the Author

1. Is the manuscript technically sound, and do the data support the conclusions?

Reviewer #1: No

Reviewer #2: Yes

2. Has the statistical analysis been performed appropriately and rigorously?

Reviewer #1: No

Reviewer #2: Yes

3. Have the authors made all data underlying the findings in their manuscript fully available?

Reviewer #1: No

The data are available upon request and can currently be found at i10.6084/m9.figshare.22111211. 

Reviewer #2: Yes

4. Is the manuscript presented in an intelligible fashion and written in standard English?

Reviewer #1: Yes

Reviewer #2: Yes

5. Review Comments to the Author

Reviewer #1: COMMENTS

Femicide in Mexico: Evidence of an increasing trend

General remarks

1. The title of the article does not explain the topic correctly, if a statistical analysis was carried out then it should be indicated explicitly. What “evidence” of an increasing trend has been found?

Answer 1: We agree, the title was changed accordingly to ¨Femicide in Mexico: Statistical evidence of an increasing trend¨.

2. The definition of crimes against life should be pointed out explicitly in the abstract. Against which other crimes are femicides compared to in the article?

Answer 2: In the new version, we point out in the abstract that “Life and bodily integrity crimes include homicide, culpable homicide, injuries, malicious injuries, abortion and other crimes that threaten life.” The same is explained in more detail in page 9 before Table 1. Descriptive Statistics: “During the analyzed period, 13,814,735 crimes were registered, out of these 1,809,735 cases (13.10%) belong to the Life and bodily integrity crimes (homicide, culpable homicide, injuries, malicious injuries, abortion and other crimes that threaten life). In this juridical type of crime, 5,759 femicide cases were registered (with StDev 2.66), which represents 66.20 registered cases per month (per 100,000 habitants) in the whole country. Table 1 summarizes the descriptive statistics of the registered crimes for all 32 Mexican states”.

3. The phrase “We understand that all femicides are not reported or that there are cases that are not adequately classified” is important, since definitely this is a problem. Why point out that femicides are not adequately reported or classified? Better definitions or doing research on the topic is definitely needed at this point.

Answer 3: The term “femicide” has only been recognized in 2015 in Mexico. Some portion of the increase in femicides may be due to increased recognition of the phenomenon, but this cannot explain the overall increasing trend or the difference we observe between crimes against life trends and femicide trends. Thus, we point out that femicides are still not adequately reported or classified because it is very likely that femicide rates are in fact even higher than our results indicate. This is very important to set the course for future studies to examine the real magnitude of the problem.

4. In general, it is interesting to compare data of different periods of time. This should be treated carefully though, since temporal data of femicides in Mexico is very scattered across time; because of this fact we suggest revising the whole aim of the paper once again.

Answer 4: Yes, we agree that the data has limitations, and we acknowledge this in the paper. However, official statistics are less available for femicide than other violent phenomena. For this reason, we are using openly available data to compare femicide with crimes against life (e.g. involuntary and culpable homicide, injuries, and other crimes that threaten life) to determine whether or not femicide is growing in Mexico. Acknowledging that there is a separate growing trend of femicides can set the base for collection of better sets of data in future that can measure the magnitude of the problem. Despite data limitations, our study is not the first in the literature to conduct trend studies with this type of data. Recently, for example, Karakasi et al. (2022) carried out a similar study in Greece and Acosta (2015) used Mexican data to show an increasing trend of femicides in Mexico but without statistical evidence. Additionally, Frias (2023) questions the presence of an increase in femicides in Mexico using data from 2010-2017. 

Mistakes in methodology, writing and numbering mistakes

1.Throughout the paper, assumptions of normality and independence are made in the data, but this is not well supported throughout the text. It is necessary to establish clearly why it can be thought that the phenomenon complies with normality and independence.

Answer 1: We based this assumption of normality in the Central Limit Theorem (CLT). CLT is a statistical premise that, given a sufficiently large sample size (more than 30) from a population with a finite level of variance, the mean of all sampled variables from the same population will be approximately equal to the mean of the whole population. Furthermore, these samples approximate a normal distribution, with their variances being approximately equal to the variance of the population as the sample size gets larger, according to the law of large numbers. In the case of the comparison of the states we have 32 states with 87 periods for each sample.

Although we assume normality, it is difficult to explain the non-independence and the equality of the variances between the samples so we perform a normality test on each of the samples. The results did not comply with normality, so we switched to nonparametric tests. The same results are obtained from the nonparametric tests as with ANOVA: the number of femicides between states is statistically significant.

We also explained the normality assumption in the proportion test in page 14

“According to the CLT, when n is large and you want to calculate a probability like P(a<X<b) all that is required is to assume that the mean is normal and standardize it, the response will be approximately normal (Devore, 2016)”.

2. Page 5 the authors specify the numbering of the sections.

Answer 2: The body formatting guidelines of the journal doesn´t allow numbering but we used the level headings for major, section and subsections headings.

Descriptive statistics were placed in the corresponding place as follows:

The rest of the article is organized as follows. In materials and methods: the data, method, and fundamental concepts are presented. In the analysis, the results were explained considering descriptive statistics and non-parametric means comparison; also, a hypothesis proportion test is performed between the years and a forecast to describe the trend; at the end of the section the discussion is displayed. Finally, the conclusion and further research is described.

However, this numbering is not present in the titles, it is necessary to put the numbering of the sections and subsections if they exist.

3. On page 5:

Table 1 summarizes the descriptive statistics of the registered crimes for all 32 Mexican states.

This corresponds to a result. The descriptive analysis is presented in the Method section in point 1. This table should appear in the results and not in the Data section.

Answer 3: This was corrected, descriptive statistics are now in page 4 in Method section and in Analysis and discussion in page 8

4. The results section begins with the ANOVA results, this corresponds to point 2 (page 6) of the three stages of the analysis. This section should begin with point 1 of the analysis stages (Descriptive statistics).

Answer 4: This was corrected, descriptive statistics are now in page 8 in Analysis and discussion. 

5. The graphs should be improved, for example on page 16 in one of the graphs the years appear at the bottom and in the other at the top. It must standardize the presentation of the graphs and improve the quality.

Answer 5: The graphs were improved, and we standardized them to improve the quality.

6. The forecast part does not appear in the analysis stages nor in the methods section, it should be presented clearly in the methods, what double exponential smoothing consists of as they did with the other methods.

Answer 6: This was corrected; forecasting methods is in page 7 in the methods sections and in page 16 in the analysis.

7. On page 13, the tables to which it refers must be 4 and 5, since they are the ones that describe the statistical significance of the number of femicides.

“In other words, while in Tables 4 and 5 there are statistically significant changes in the number of femicides, in the total number of life and bodily integrity crimes no statistically significant trend is observed”. In the case of all crimes against life and bodily integrity, if it corresponds to tables 6 and 7.

Answer 7: Thank you for the comment, this was corrected as follows: 

We analyze whether a similar trend can be observed in the case of the reported numbers of life and bodily integrity crimes. In Tables 6 and 7, the same tests were applied. In this case, the in the Kruskal-Wallis and Mood median tests, the p-value is greater than 0.05 (p=0.097 and p=0.070 respectively). Therefore, H_0 is rejected, that is, there are no statistically significant differences between the years in life and bodily integrity crimes. In other words, while in Tables 4 and 5 there are statistically significant changes in the number of femicides, in the total number of life and bodily integrity crimes no statistically significant trend is observed.

.

1. In the ANOVA analysis, the variability of the behavior of femicides in the period of time is very large, if we take the average 2.0692 cases of femicides per 100,000 inhabitants and the standard deviation 1.9265, we can verify the great variation in the data by calculating the coefficient of variation of 93%.

If we analyze the columns of the mean and the standard deviation of Table 2 (page 10), we observe some states with not very high coefficients of variation, which were listed with a high number of femicides, except for the state of Morelos:

State Mean StDev Cv

Ciudad de México 4.786 2.742 57.3

Jalisco 4.464 2.595 58.13

Estado de México 8.548 4.433 51.8

Morelos 1.1711 2.345 49.94

Veracruz 6.524 3.198 49.02

Answer : The variability of states is not so high with respect to the others, so it was necessary to evaluate whether these states met the normal conditions. If the assumption of normality was fulfilled, the femicides in these states would be intrinsic natural variabilities, that is, characteristics of the states. In the other states, the dynamics would be non-linear to the presence of femicides. We evaluated the normal conditions in S2 Fig , the ANOVA analysis was replaced with a nonparametric test because the assumption of normality was not fulfilled.

General evaluation

Despite the fact that the topic is very important, the paper is in many ways incomplete and very poorly written.

We revised the manuscript carefully. If the reviewer has specific suggestions regarding grammar or structure, we would be happy to adjust the manuscript accordingly. 

Reviewer #2: The manuscript analyzes an issue of special interest in Mexico: whether the practice of femicide has increased compared to other equivalent crimes. The study is based on official information from the Executive Secretariat of the National Public Security System from which the evidence presented is drawn. Although the evidence supports the hypothesis that femicides have increased in the country, it is considered relevant for the co-authors to review their analytical framework, specifically to reflect on the paradox of information in human rights (Sikkink), a situation that occurs when a practice is named and begins to be counted. While the evidence presented is compelling, it is important to develop reasons why their analysis would go some way to controlling the information paradox. On the other hand, given that it does not make the central argument of the paper, it is suggested to avoid the reference to the relationship between the approval of legislation and the increase in crime as well as to the macho culture.

We appreciate this comment and, in accordance with the reviewer’s suggestion, deleted the mentions of novel legislation. However, we feel that the reference to “machismo” is warranted in that it helps set the context for readers unfamiliar with the prevalence or level of acceptance of domestic abuse and violence against women in Mexico and do not see how machismo relates to the information paradox. 

6. PLOS authors have the option to publish the peer review history of their article (what does this mean?). If published, this will include your full peer review and any attached files.

Do you want your identity to be public for this peer review? For information about this choice, including consent withdrawal, please see our Privacy Policy.

Reviewer #1: No

Reviewer #2: No

---

## [Decision Letter · Decision Letter 1]

3 Aug 2023

Femicide in Mexico: Statistical Evidence of an Increasing Trend

PONE-D-22-26945R1

Dear Dr. Flegl,

We’re pleased to inform you that your manuscript has been judged scientifically suitable for publication and will be formally accepted for publication once it meets all outstanding technical requirements.

Kind regards,

Jesús Espinal-Enríquez

Academic Editor

PLOS ONE

Additional Editor Comments (optional):

Dear Dr. Flegl.

Thank you for submitting your manuscript to PLoS ONE. I extend my apologies for the delay in the revision process.

Having thoroughly reviewed the comments provided by Reviewer #1, along with my own assessment, I am pleased to inform you that your manuscript has been accepted for publication. I believe this piece of work is comprehensive and highly valuable for policy makers, shedding light on the pressing issue of femicide.

Once again, I appreciate your submission and look forward to reading further complementary research from you.

Best regards,

Reviewers' comments:

Reviewer's Responses to Questions

**Comments to the Author**

1. If the authors have adequately addressed your comments raised in a previous round of review and you feel that this manuscript is now acceptable for publication, you may indicate that here to bypass the “Comments to the Author” section, enter your conflict of interest statement in the “Confidential to Editor” section, and submit your "Accept" recommendation.

Reviewer #2: All comments have been addressed

2. Is the manuscript technically sound, and do the data support the conclusions?

Reviewer #2: Yes

3. Has the statistical analysis been performed appropriately and rigorously? 

Reviewer #2: Yes

4. Have the authors made all data underlying the findings in their manuscript fully available?

Reviewer #2: Yes

5. Is the manuscript presented in an intelligible fashion and written in standard English?

Reviewer #2: Yes

6. Review Comments to the Author

Reviewer #2: I consider the authors fulfilled the comments I originally made. Taking into account my area of expertise, the conceptual clarification I did was answered adecuatelly.

7. PLOS authors have the option to publish the peer review history of their article (what does this mean?). If published, this will include your full peer review and any attached files.

Reviewer #2: No

---

## [Editor Report · Acceptance letter]

8 Aug 2023

PONE-D-22-26945R1 

Femicide in Mexico: Statistical Evidence of an Increasing Trend 

Dear Dr. Flegl:

I'm pleased to inform you that your manuscript has been deemed suitable for publication in PLOS ONE. Congratulations! Your manuscript is now with our production department. 

Kind regards, 

on behalf of

Dr. Jesús Espinal-Enríquez 

Academic Editor

PLOS ONE